# Evading Overlapping Community Detection via Proxy Node Injection

## Abstract

Protecting privacy in social graphs requires preventing sensitive information, such as community affiliations, from being inferred by graph analysis, without substantially altering the graph topology. We address this through the problem of *community membership hiding* (CMH), which seeks edge modifications that cause a target node to exit its original community, regardless of the detection algorithm employed. Prior work has focused on non-overlapping community detection, where trivial strategies often suffice, but real-world graphs are better modeled by overlapping communities, where such strategies fail. To the best of our knowledge, we are the first to formalize and address CMH in this setting. In this work, we propose a deep reinforcement learning (DRL) approach that learns effective modification policies, including the use of proxy nodes, while preserving graph structure. Experiments on real-world datasets show that our method significantly outperforms existing baselines in both effectiveness and efficiency, offering a principled tool for privacy-preserving graph modification with overlapping communities.

## 1 Introduction

Graphs often exhibit rich community structure, where nodes organize into densely connected groups that capture meaningful relations (Girvan & Newman, 2002a). Detecting such communities is a fundamental task with broad applicability: in biology, it reveals functional modules in protein–protein interaction networks; in network security, it helps identify clusters of coordinated malicious actors; and in social networks, it identifies tightly knit circles of users. Because of its ability to expose latent organization in complex systems, community detection (Blondel et al., 2008) has become a cornerstone of graph analysis and an essential tool across multiple domains (Javed et al., 2018).

However, the widespread use of community detection also raises privacy concerns. In social graphs, revealing a user's community membership can expose sensitive attributes, such as political affiliation, health condition, or social group identity, even when the user has not explicitly disclosed this information. For instance, a user who frequently interacts with members of a sensitive community may be algorithmically inferred to share that affiliation, exposing private beliefs and making them vulnerable to profiling or discrimination.

To address this risk, we study the problem of community membership hiding (CMH), where the goal is to alter a graph's edge structure so that a target node is no longer identified as part of its original community, regardless of the detection algorithm. Prior work (Bernini et al., 2024; Silvestri et al., 2025) has focused mainly on non-overlapping community detection, where each node belongs to a single community. In this setting, the problem admits trivial solutions – for instance, attaching the target node to a random graph effectively obscures its membership. In contrast, under *overlapping community detection*, such strategies fail: since nodes can legitimately belong to multiple communities, disentangling the target node from its original community becomes a non-trivial challenge.

To the best of our knowledge, we are the first to formally study community membership hiding (CMH) under this more realistic and challenging scenario. We propose a deep reinforcement learning (DRL) approach that introduces a small set of external *proxy nodes*, initially organized as an Erdős-Rényi graph and connected to the target node. A DRL agent then learns an optimal policy for edge modifications, either by rewiring the target's connections or manipulating links between proxies and the rest of the graph. This strategy achieves effective membership hiding while minimizing disruption to the overall graph topology.

To summarize, our main contributions are:

(1) We formally define the CMH problem under overlapping community detection and show that trivial strategies succeed in the non-overlapping case, but fail in this more realistic setting, motivating the need for a principled solution;

(2) We propose a DRL-based framework that learns optimal edge modification policies for both the target node and injected proxy nodes;

(3) We demonstrate on real-world graph datasets that our method significantly outperforms existing baselines. Source code is available at: `https://anonymous.4open.science/r/overlapping_comm_detect-EC02/README.md`.

The paper is organized as follows. Section 2 reviews related work, and Section 3 introduces background and notation. In Section 4, we formally define the CMH problem in both non-overlapping and overlapping settings, and analyze the impact of proxy injections in each case. Section 5 presents our proposed approach for the overlapping scenario, followed by experimental results in Section 6. Section 7 discusses limitations and future directions, and Section 8 concludes the paper.

## 2 RELATED WORK

Community detection has long been a central theme in graph analysis, with algorithms broadly divided into two categories: non-overlapping methods, such as modularity maximization (Blondel et al., 2008), spectral clustering (Ruan et al., 2012), random walk, label propagation, or statistical inference (Airoldi et al., 2008), which partition the graph into disjoint sets; and overlapping methods, such as clique percolation, label propagation–based techniques (Rossetti, 2020; Coscia et al., 2012), matrix factorization (Yang & Leskovec, 2013), Neighborhood-Inflated Seed Expansion (Whang et al., 2015), or techniques based on minimizing the Hamiltonian of the Potts model (Ronhovde & Nussinov, 2009), which allows nodes to belong to multiple communities simultaneously and better reflects the eclectic structure of real-world social graphs.

The problem of community membership hiding (CMH) was first introduced by Bernini et al. (2024), who proposed modifying a node's local neighborhood to obscure its community affiliation. They formulate the task as a counterfactual graph optimization problem, using a graph neural network (GNN) to model the structural dependencies and a deep reinforcement learning (DRL) agent within a Markov decision process framework to rewire edges under a strict budget, ensuring efficiency and realistic perturbations. More recently, Silvestri et al. (2025) reformulated CMH as a differentiable optimization problem, inspired by adversarial attacks on GNNs. Their method perturbs the target's adjacency vector via a learnable perturbation and relies on gradient-based optimization guided by "promising actions", a privileged set of candidate modifications based on structural properties. Crucially, both works restrict attention to non-overlapping communities, where trivial solutions (e.g., attaching the target to an external graph) often suffice (see Section 4.3). By contrast, we address the more realistic and challenging overlapping setting, where community boundaries are less rigid and hiding membership requires more nuanced strategies.

While strict CMH has primarily been studied in non-overlapping contexts, recent works have investigated privacy preservation in overlapping settings under distinct formulations (Liu et al., 2022; Yang et al., 2022; Han et al., 2025). These approaches typically focus on *overlapping area avoidance*, forcing a target node out of intersection sets into a single community, or rely on maximizing differences in discrete membership probabilities. A fundamental limitation is their reliance on discrete metrics (e.g., community counts or binary flags), which fail to capture the dynamic nature of community detection. For instance, if a detector generates a new community ID with nearly identical member composition, discrete methods would erroneously classify this as successful hiding. In contrast, we define success via rigorous set-similarity metrics, aiming to hide specific sensitive affiliations regardless of whether the node remains in an overlapping region or joins new communities.

Our approach draws inspiration from recent node-injection strategies, which show that introducing proxy nodes can effectively mislead GNN-based node classifiers on specific targets while requiring only minimal perturbations to the graph (Wang et al., 2021; 2018).

Beyond CMH, our work connects to the broader literature on adversarial attacks in graph learning, which exposes vulnerabilities in tasks such as link prediction (Chen et al., 2020), node and graph

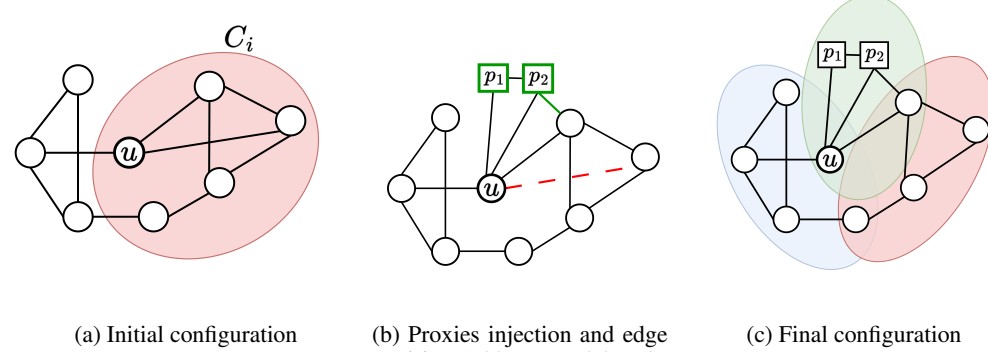

(a) Initial configuration          (b) Proxies injection and edge          (c) Final configuration
                                       rewiring (add=green, del=red).

Figure 1: Example of the proxy-based CMH problem in the overlapping community detection setting for budget of modifications $\beta = 2$, and number of proxies $k = 2$.

classification (Dai et al., 2018), and graph clustering (Chen et al., 2017). Within clustering, the most related direction is community deception (Fionda & Pirro, 2017; Waniek et al., 2018b; Chen et al., 2019), where the objective is to perturb the graph to obscure an entire community. Our focus differs in both granularity and motivation: instead of hiding a whole group, we tackle the fine-grained challenge of concealing the community membership of a single target node, a problem that is both more subtle and directly tied to user privacy.

## 3 PRELIMINARIES

**Notation** Let $G = (V, E)$ be an undirected[1] graph, where $V$ is the set of nodes with $|V| = n$, and $E \subseteq V \times V$ is the set of edges with $|E| = m$. Throughout, we use $sim(\cdot, \cdot)$ to denote a similarity function between sets, and $\tau \in (0, 1)$ to denote a similarity threshold.

**Community Detection** Community detection aims to partition the nodes of a graph into clusters, or communities, with denser internal connectivity than external links. In this work, we focus on structure-based methods, leaving node-attribute-based methods for future work. Formally, a community detection algorithm can be described as a function $f(\cdot)$ that maps a graph $G$ to a set of nonempty communities $f(G) = \{C_1, C_2, \ldots, C_k\}$, where $k$ is not fixed a priori. For a node $u \in V$, $f(u, G)$ denotes the set of communities containing $u$. In the non-overlapping setting, $|f(u, G)| = 1$ for all $u \in V$, whereas in the overlapping setting, there exists at least one $u$ with $|f(u, G)| \neq 1$.[2] Notably, the input to $f$ may range from the adjacency matrix $\boldsymbol{A}$ to richer representations such as $Z = \Phi_\eta(\boldsymbol{A}, \boldsymbol{X})$, where $\boldsymbol{X}$ is the node feature matrix and $\Phi_\eta$ is a GNN encoder. For simplicity, we denote the generic input representation as $G$.

**Detection Algorithms** In the non-overlapping setting, used to motivate our proxy injection strategy in Section 4.3, we follow the protocol of Bernini et al. (2024) and select three algorithms that cover distinct topological approaches. We employ `greedy` (Brandes et al., 2008) and `louvain` (Blondel et al., 2008), which both detect communities by optimizing the global modularity function via hierarchical agglomeration, and `walktrap` (Pons & Latapy, 2005), which infers community structure based on random walks. For the overlapping setting, which is the primary focus of this study, we select `angel` (Rossetti, 2020) and `demon` (Coscia et al., 2012). Unlike partition-based methods, these algorithms operate on a "local-first" principle, extracting dense structures from local neighborhoods or ego-networks and subsequently merging them based on an overlap threshold $\phi$, thereby allowing nodes to naturally hold multiple memberships. We set the merging threshold $\phi = 0.8$ for both algorithms based on empirical observations, as this value yielded the most mean-

---

[1]The reasoning naturally extends to directed graphs as well.

[2]While overlapping detection algorithms allow nodes to remain unassigned (i.e., $|f(u, G)| = 0$), we exclude such nodes from our experiments to avoid selection bias and ensure a non-vacuous hiding objective.

ingful community structures across our datasets (see Table 4 and Appendix B). For more details about the detection algorithms, refer to Appendix A.

# 4 COMMUNITY MEMBERSHIP HIDING

At a high level, the community membership hiding (CMH) problem involves perturbing the edge structure of a graph to conceal the association of a target node with a specific community $C_i$, while preserving the global topology as much as possible. Formally, following Bernini et al. (2024), the goal is to learn an optimal policy $h_\theta$ that transforms the original graph $G$ into a modified graph $G' = (V, E')$ such that, when the detection algorithm $f(\cdot)$ is applied to $G'$, the target node is no longer assigned to $C_i$. We model $f(\cdot)$ as a black box, relying solely on its input–output mapping.

The notion of a node being "hidden" from a community is inherently task-dependent. One may, for instance, require the target to share no memberships with a given set of nodes, or simply to reduce its similarity to its original community. Following prior work, we adopt the latter view, using a similarity function between sets $sim(\cdot, \cdot)$ and a threshold $\tau \in [0, 1)$.

We analyze the problem under two settings: the *non-overlapping* case, where communities are disjoint and can be addressed by trivial strategies, and the more general *overlapping* case, where nodes may belong to multiple communities and the problem remains non-trivial.

## 4.1 NON-OVERLAPPING SCENARIO

In the non-overlapping setting, communities form a partition of the node set $V$. Following previous work (Bernini et al., 2024; Silvestri et al., 2025), the hiding objective is defined via a threshold-based similarity criterion. Let $C_i = f(u, G)$ denote the target node $u$'s community in the original graph, and let $\{C'_1, \ldots, C'_{k'}\}$ be the communities detected in the perturbed graph $G'$, with $u$ assigned to the community $C'_i$. The hiding task is successful if the similarity between the original and updated community (excluding $u$) falls below the threshold $\tau$, i.e. $\mathrm{sim}(C_i \setminus \{u\}, C'_i \setminus \{u\}) \leq \tau$.

Given the complexity of the problem, prior work restricts the search to solutions within a modification budget $\beta$. Effectiveness is then evaluated using the F1 score, defined as the harmonic mean of *success rate* (SR), i.e., the proportion of trials in which the hiding condition is satisfied, and *normalized mutual information* (NMI) (Strehl & Ghosh, 2002): SR measures the ability to hide the target node, while NMI quantifies how well the original community structure is preserved. Their combination offers a balanced view of both hiding effectiveness and minimal disruption.

## 4.2 OVERLAPPING SCENARIO

In many real-world graphs, most notably social graphs, nodes often participate in multiple groups. Overlapping community detection methods capture this by allowing nodes to belong to several communities simultaneously, posing a more nuanced and practically relevant challenge for CMH.

To extend the formulation, let $C_i \in f(u, G)$ be the target community of node $u$ in the original graph. The hiding condition is satisfied if, for *every* community $C'_i$ assigned to $u$ in the perturbed graph $G'$, the similarity with $C_i$ falls below the threshold $\tau$:

$$\mathrm{sim}(C_i \setminus \{u\}, C'_i \setminus \{u\}) \leq \tau, \quad \forall\, C'_i \in f(u, G'). \tag{1}$$

If $u$ is not assigned to any community in $G'$ (i.e., $f(u, G') = \emptyset$), the condition holds vacuously, representing a successful hiding attempt since the detector cannot place $u$ in any group. To avoid bias, we exclude from our experiments nodes that are not initially assigned to any community.

## 4.3 MOTIVATION: THE FAILURE OF NAÏVE INJECTIONS

A recurring insight in adversarial graph learning is that injecting new nodes can substantially alter model behavior while leaving the overall structure largely intact (Wang et al., 2021; 2018). Motivated by this, we extend the CMH problem to allow the target node to introduce a small set of controllable *proxy* nodes. This assumption is realistic in social graphs, where members can create new nodes independent of the original topology and strategically connect them to pursue their goal.

Formally, given a proxy set $P = \{p_1, \ldots, p_k\}$ with $k \ll |V|$, we generate an Erdős–Rényi graph $G_{proxy} \sim G_{ER}(k, p)$, where $p$ is the edge probability, and connect all proxies to the target node $u$. We refer to this strategy as the *naïve connection* baseline. In the following analysis, we evaluate this approach to demonstrate that while such generic perturbations are effective for non-overlapping detection, they fail to obscure membership in the overlapping setting.

**Datasets**   We ground this study in real-world graphs from social, linguistic, and collaboration domains. For non-overlapping experiments (Section 4.3), we use five benchmarks: `kar`[3], Zachary's karate club; `words`[3], capturing semantic associations between common English words; `vote`[4], which records Wikipedia administrator elections and the votes cast between users; `fb-75`[3], a Facebook ego network; and `pow`[3], representing the American power grid. For overlapping experiments, we retain the same set, except that `fb-75` is replaced with `nets`[5], a coauthorship graph in network science, since the available implementation of overlapping community detection algorithms scales poorly to large graphs. Indeed, even the preliminary experiments of Section 4.2 on `fb-75` could not be completed within a reasonable timeframe. Detailed dataset statistics are reported in Table 1.

Table 1: Dataset properties and number of communities as detected by each of our community detection algorithms.

| Graph Properties | | | Non-overlapping $f(\cdot)$ | | | Overlapping $f(\cdot)$ | |
|---|---|---|---|---|---|---|---|
| Dataset | $|V|$ | $|E|$ | greedy | louvain | walktrap | demon | angel |
| kar | 34 | 78 | 3 | 4 | 5 | 5 | 2 |
| words | 112 | 425 | 7 | 7 | 25 | 16 | 4 |
| vote | 889 | 2,900 | 12 | 10 | 42 | 188 | 161 |
| pow | 4,941 | 6,594 | 40 | 41 | 364 | 99 | 30 |
| nets | 1,589 | 2,742 | 403 | 405 | 416 | 112 | 120 |
| fb-75 | 6,386 | 217,662 | 29 | 19 | 357 | 5,551 | 48 |

**Non-overlapping Results**   Figure 2 shows the resulting F1 scores across datasets and non-overlapping detection algorithms. To ensure comparability with prior work, we set the number of proxies to match the modification budget $\beta$, defining $\mu = |E|/|V|$ and varying $k$ over $0.5\mu$, $\mu$, and $2\mu$. Notably, even without explicit edge rewiring from the target node, the results remain consistently close to, and in several cases even surpass, the DRL approach of Bernini et al. (2024) (red reference lines). The sole exception is the `pow` dataset, where the grid-like structure limits the effectiveness of the naïve connection strategy. Overall, these findings underscore the utility of proxy nodes for concealing community membership and motivate their use as a powerful mechanism for CMH.

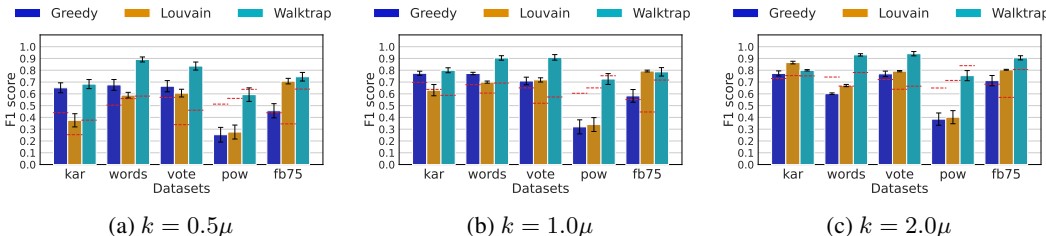

(a) $k = 0.5\mu$     (b) $k = 1.0\mu$     (c) $k = 2.0\mu$

Figure 2: F1 scores of SR and NMI for the naïve connection baseline under non-overlapping detection algorithms, evaluated across varying proxy budgets $k$ (with $\tau = 0.5$, $p = 0.5$). Red dashed lines indicate the DRL-Agent performance for each setting, as reported by Bernini et al. (2024)

---

[3] http://konect.cc/

[4] https://networkrepository.com

[5] http://www-personal.umich.edu/~{}mejn/netdata

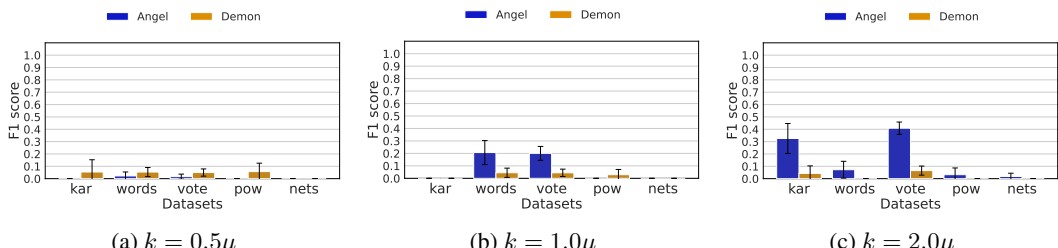

(a) $k = 0.5\mu$      (b) $k = 1.0\mu$      (c) $k = 2.0\mu$

Figure 3: F1 scores of SR and ONMI for the naïve connection baseline under overlapping community detection algorithms, evaluated across varying proxy budgets $k$ (with $\tau = 0.5$, $p = 0.5$).

**Overlapping Results**   Figure 3 reveals a critical limitation: the naïve connection baseline, that proved effective for non-overlapping detection, fails precipitously under the overlapping setting. This performance gap stems from a fundamental difference in detection logic. While non-overlapping algorithms are forced to choose a single partition, effectively "pulling" the target into the proxy cluster, overlapping algorithms can simply assign the target to *both* the original community and the new proxy group. Consequently, the target is not hidden but merely multi-affiliated.

This finding underscores the need for a refined policy that does not simply add external distractions but actively undermines the specific structural features that tie the target to its target community.

## 5   PROPOSED METHOD: ODRL AGENT

Building on the formulation of Bernini et al. (2024), we model the community membership hiding (CMH) problem as a discounted Markov decision process (MDP) $\mathcal{M} = (S, \mathcal{A}, \mathcal{P}, r, \gamma)$. In this setting, the state $s_t$ corresponds to the current graph $G_t$, and the transition function $\mathcal{P} : S \times \mathcal{A} \times S \to [0, 1]$ is deterministic: $P(s_{t+1}|s_t, a_t) = 1$ if $s_{t+1}$ is the graph obtained by applying action $a_t$ to $s_t$, and 0 otherwise. The single-step action space for the controlled node $u$ consists of (i) deletions of any existing edge incident to $u$, and (ii) additions of edges between $u$ and any non-neighboring node $v \in V$:

$$\mathcal{A}_u = \{\text{del}(u, v) \mid (u, v) \in E\} \cup \{\text{add}(u, v) \mid (u, v) \notin E\} \tag{2}$$

Therefore, the complete set of single-step candidate actions is given by $\mathcal{A} = \bigcup_{v \in \{u\} \cup P} \mathcal{A}_u$, i.e., the union of all the single-step action spaces for each controlled node. This formulation contrasts with Bernini et al. (2024), who restrict modifications based on the original community of the target node. By removing this constraint, our approach enhances the agent's expressiveness, allowing for the exploration of a broader range of intermediate graph configurations.

The reward associated with taking action $a_t$ in state $s_t$, leading to the next state $s_{t+1}$, is defined as

$$r(s_t, a_t) = \begin{cases} 1 + \alpha \delta^t_{sim}, & \text{if Eq. (1) is satisfied,} \\ \alpha \delta^t_{sim}, & \text{otherwise,} \end{cases} \tag{3}$$

where $\delta^t_{sim} = \frac{\text{sim}_{prev} - \text{sim}_{curr}}{\text{sim}_{prev}}$ measures the relative decrease in the maximum similarity between the original target community and the community assignment at time $t$, normalized by the previous similarity score. In our experiments, we set $\alpha = 0.1$.

**Factored Policy**   A naïve policy $\pi_\theta$ that scores all ordered node pairs $(i, j)$ would require the computation of $O(|V|^2)$ logits per episode step, which is prohibitive even for graphs of intermediate size. Instead, inspired by similar methods on off-policy algorithms (Tavakoli et al., 2018), we designed a factored action space that is aligned with the problem structure by $(i)$ select which acting node $u_j \in \{u\} \cup P$ will perform the edit and $(ii)$ conditionally select an edit for $u_j$ by jointly selecting the target node and whether to perform an addition or removal. Formally, we have:

$$\pi_\theta(a_t|s_t) = \pi_\theta^{node}(j|s_t)\pi_\theta^{actor}(a|s_t, j) \tag{4}$$

For each actor branch of the policy, we perform action masking such that $\pi_\theta^{actor}(a|s_t, j) = 0$ if $a \notin A_{u_j}$, reducing variance during learning and preventing invalid operations. With this formulation, each actor branch only has to score $2(|V| - 1)$ logits, for a total computation of $O(|\{u\} \cup P||V|)$ logits. Knowing that $|\{u\} \cup P| \ll |V|$, the space complexity of our policy becomes $O(|V|)$.

**Training** We learn the optimal policy $\pi_\theta$ using Proximal Policy Optimization (PPO) (Schulman et al., 2017). The complete training objective combines the standard PPO clipped surrogate $\mathcal{L}_{clip}$ loss with a value-function loss $\mathcal{L}_{value}$ and an entropy bonus to encourage exploration:

$$\mathcal{L}(\theta) = c_v \mathcal{L}_{value} + c_{clip} \mathcal{L}_{clip} - c_{ent} \mathbb{E}_t \left[ \mathcal{H}(\pi(\cdot | s_t)) \right], \tag{5}$$

where $c_v$, $c_{clip}$, and $c_{ent}$ are weighting coefficients.

A key aspect of our approach is adapting this objective for our factored action space. Since the policy is divided into selecting an acting node and then an edge modification, we compute the policy probability ratios for each factor independently. For a sampled action $a = (i, a_{actor})$, these are:

$$r_t^{\text{node}} = \frac{\pi_\theta^{\text{node}}(i \mid s_t)}{\pi_{\theta_{\text{old}}}^{\text{node}}(i \mid s_t)}, \qquad r_t^{\text{actor}} = \frac{\pi_\theta^{\text{actor}}(a_{\text{actor}} \mid s_t, i)}{\pi_{\theta_{\text{old}}}^{\text{actor}}(a_{\text{actor}} \mid s_t, i)}, \tag{6}$$

The PPO clipping is applied to each ratio, and the final surrogate objective is the average of the two resulting losses: $\mathcal{L}_{clip}(\theta) = \frac{1}{2}(\mathcal{L}_{node}(\theta) + \mathcal{L}_{actor}(\theta))$. Ultimately, the training objective can be summarized as:

$$\theta^* = \arg \min_\theta \left\{ \mathcal{L}(\theta; u, f, G_{proxy}, G) \right\}$$
$$\text{subject to: } |\mathcal{B}_u| \leq \beta, \tag{7}$$

where $\mathcal{B}_u \subset \mathcal{A}_u$ denotes the set of edge modifications performed by the actors, i.e., $\{u\} \cup P$.

**Actor and Critic Networks** Our agent's architecture uses a shared graph encoder to generate rich, $d_h$-dimensional node embeddings. This encoder is a 4-layer GCN (Kipf & Welling, 2016) with jumping knowledge connections (Xu et al., 2018) to capture multi-scale structural patterns. From these embeddings, we aggregate representations for the target node, the proxy nodes, and the entire graph. This aggregated context is then processed by a GRU (Cho et al., 2014) to produce a temporal state representation, $h_{\text{shared}}$, which serves as the input for our actor and critic heads.

The actor employs a factored policy that first selects an acting node from the $|P| + 1$ candidates (the target and its proxies), and then conditionally selects an edge modification from the $2(|V| - 1)$ available actions for that node. In parallel, a simple two-layer MLP critic estimates the state value directly from $h_{\text{shared}}$. More details are available in Appendix B.

## 6 Experiments

### 6.1 Experimental Setup

**Similarity and Evaluation Metrics** Following prior work (Bernini et al., 2024; Silvestri et al., 2025), we adopt the Sørensen–Dice coefficient (Dice, 1945) as the similarity function $sim(\cdot, \cdot)$. We assess concealment performance by considering both the *Success Rate* (SR), i.e., the fraction of trials where the hiding condition in Eq. (1) is met, and the *Overlapping Normalized Mutual Information* (ONMI) (McDaid et al., 2011), which quantifies the similarity between community assignments before and after perturbations. Since SR and ONMI are contrasting metrics, we also compute their harmonic mean (F1 score), computed as $\frac{2 \times \text{SR} \times \text{ONMI}}{\text{SR} + \text{ONMI}}$.

**Baselines** We evaluate our proposed method, referred to as *ODRL* agent, against four baseline strategies. All baselines operate on the same graph structure with injected proxy nodes but employ different edge-modification policies. The *Random-based* baseline selects one actor (target or proxy) and one endpoint node uniformly at random at each step and toggles the edge between them. The *Degree-based* heuristic modifies the connection between the actor and the endpoint nodes with the highest degree. Similarly, the *Centrality-based* approach selects the actor and endpoint for the edge

modification based on the highest betweenness centrality (Freeman, 1977). Finally, our *Roam-based* baseline adapts the ROAM heuristic (Waniek et al., 2018a); it first removes the edge between the target node and its highest-degree neighbor, and then reconnects that neighbor to different neighbors of the target node.

**Experimental Design** We follow the evaluation protocol of Silvestri et al. (2025), sampling 100 nodes from communities whose sizes are $0.3, 0.5, 0.8$ times that of the largest community and restricting inter-community similarity so that $\text{sim}(C_i, C_j) \leq 0.8$. To ensure statistical reliability, we sample three communities for each size. The ODRL agent is trained only against the `angel` detector. We then test its performance in two settings: a symmetric setting, testing against `angel`, and a more realistic asymmetric one, testing against `demon`, an unseen detector. Strong performance in the asymmetric setting highlights our model's *transferability* and robustness.

**Training Details** We train the ODRL agent for $5,000$ episodes using PPO optimized via RMSProp (Hinton, 2012) with a learning rate of $\eta = 5 \times 10^{-4}$. To prevent overfitting and promote policy robustness, we employ a dynamic sampling strategy that resamples the target node every 5 episodes and the target community every 50 episodes. The hidden dimension for the underlying graph encoder and policy networks is set to $d_h = 32$. We refer the reader to Appendix B for a complete listing of hyperparameters, including Generalized Advantage Estimation (GAE) (Schulman et al., 2016) settings, loss function coefficients, and entropy annealing schedules.

### 6.2 Results and Discussion

Our evaluation framework is designed to assess performance across two primary axes: the rewiring budget $\beta$ and the proxy node budget $k$. For both, we use a range relative to the average node degree, i.e., $\mu = |E|/|V|$ and $\beta, k \in \{\frac{1}{2}\mu, \mu, 2\mu\}$[6]. We comprehensively evaluate our ODRL Agent across all combinations of these budgets. The baseline methods, however, are assessed with a fixed proxy budget $k = \mu$ across the varying rewiring budgets to provide a concise point of comparison. All table entries include 95% confidence intervals (CIs). For SR, we compute CIs using the normal approximation to the binomial variance; for F1 scores, we compute 95% CIs via a non-parametric bootstrap with 1000 resamples over the sampled node set, reporting the percentile-based interval.

In the symmetric setting, our evaluation shows that the ODRL agent consistently outperforms all baselines across datasets, as reported in Table 2. The performance gap is particularly pronounced on large-scale graphs (see Figure 4a), where our method achieves substantially higher F1 scores. This trend highlights a key advantage of our approach: while baseline methods degrade as graph size increases, the ODRL agent maintains high effectiveness, demonstrating scalability and robustness.

To assess the *transferability* of our approach, we further evaluate it in the asymmetric setting, where the agent is trained against `angel` and tested against the unseen detector `demon`. As reported in Table 3, the ODRL agent generalizes effectively, outperforming the baselines across most datasets. The only exception is `words` (see Figure 4b), yet even there the performance remains competitive. Notably, the advantage of our method becomes even more evident on large-scale graphs, where the gap over baselines widens significantly. These findings underscore the practical applicability of our approach in scenarios where the adversary has no access to the target detector during training.

Lastly, we observe that the ODRL agent's performance remains stable across different numbers of proxy nodes in both symmetric and asymmetric settings. This robustness indicates that the learned policy effectively leverages available proxies without over-relying on their quantity, further validating the efficiency and adaptability of our approach. Further results, including the success rate across all settings, are provided in Appendix C.

## 7 Limitations and Future Work

While our proposed DRL-based framework effectively addresses the CMH problem for overlapping communities, we acknowledge several limitations that pave the way for future research.

---

[6]For the `kar` dataset, we set $\mu = |E|/|V| + 1$ to ensure $\beta \geq 1$.

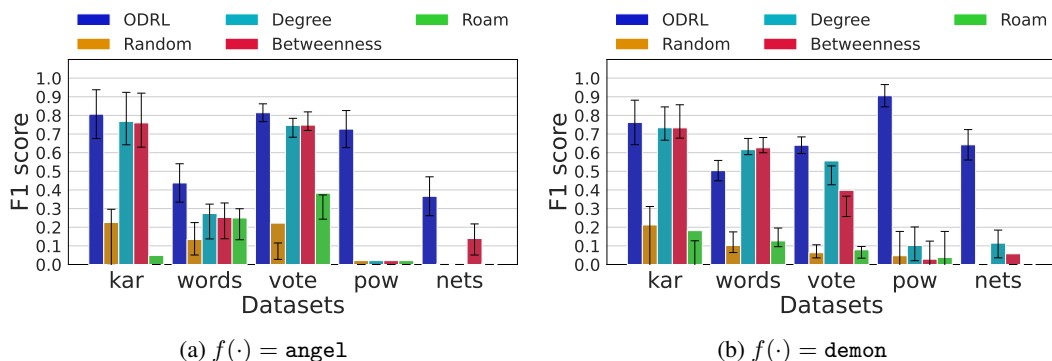

(a) $f(\cdot) = \texttt{angel}$          (b) $f(\cdot) = \texttt{demon}$

Figure 4: F1 scores of SR and ONMI for the *ODRL Agent* and baselines in the symmetric (Figure 4a) and asymmetric (Figure 4b) scenarios, with parameters $k = \mu$, $p = 0.5$, $\beta = \mu$.

Table 2: F1 Score (F1, %) $\pm$ error of SR and ONMI in the symmetric setting (angel), $\tau = 0.5$. Best results are in bold, second best are underlined

| | | Symmetric setting — training: angel; testing: angel | | | | | | |
|---|---|---|---|---|---|---|---|---|
| Dataset | $\beta$ | ODRL(0.5$\mu$) | ODRL(1.0$\mu$) | ODRL(2.0$\mu$) | Random | Degree | Betweenness | Roam |
| kar | $\frac{1}{2}\mu$ | $61.2 \pm 16.9$ | $\mathbf{71.6 \pm 14.8}$ | $56.6 \pm 17.2$ | $53.0 \pm 17.7$ | $66.0 \pm 15.9$ | $64.0 \pm 15.9$ | $23.8 \pm 16.2$ |
| | $1\mu$ | $77.7 \pm 14.4$ | $\mathbf{80.7 \pm 13.1}$ | $74.0 \pm 16.4$ | $53.0 \pm 17.7$ | $\underline{73.7 \pm 16.4}$ | $73.1 \pm 16.6$ | $7.3 \pm 8.5$ |
| | $2\mu$ | $\mathbf{90.8 \pm 11.3}$ | $\underline{90.2 \pm 11.8}$ | $85.4 \pm 16.0$ | $62.1 \pm 16.2$ | $81.1 \pm 19.1$ | $78.4 \pm 16.8$ | $7.3 \pm 8.5$ |
| words | $\frac{1}{2}\mu$ | $29.3 \pm 10.6$ | $\mathbf{34.4 \pm 11.2}$ | $32.9 \pm 10.2$ | $16.6 \pm 9.0$ | $26.1 \pm 9.1$ | $28.3 \pm 9.0$ | $26.9 \pm 8.9$ |
| | $1\mu$ | $39.0 \pm 10.5$ | $\mathbf{43.8 \pm 10.3}$ | $\underline{41.4 \pm 9.8}$ | $18.2 \pm 9.2$ | $30.3 \pm 8.8$ | $30.4 \pm 8.8$ | $30.1 \pm 8.7$ |
| | $2\mu$ | $44.6 \pm 8.6$ | $\mathbf{61.2 \pm 8.8}$ | $\underline{48.3 \pm 9.0}$ | $18.2 \pm 9.2$ | $34.0 \pm 8.5$ | $34.6 \pm 8.3$ | $34.9 \pm 7.9$ |
| vote | $\frac{1}{2}\mu$ | $\underline{81.4 \pm 4.4}$ | $79.6 \pm 4.7$ | $\mathbf{86.4 \pm 4.7}$ | $20.5 \pm 5.4$ | $61.8 \pm 4.7$ | $62.3 \pm 4.6$ | $29.4 \pm 5.7$ |
| | $1\mu$ | $\underline{83.4 \pm 4.4}$ | $81.4 \pm 4.8$ | $\mathbf{86.8 \pm 4.7}$ | $22.8 \pm 5.5$ | $77.2 \pm 4.4$ | $70.7 \pm 4.4$ | $29.4 \pm 5.7$ |
| | $2\mu$ | $\underline{84.4 \pm 4.5}$ | $81.7 \pm 4.8$ | $\mathbf{86.9 \pm 4.7}$ | $23.2 \pm 5.5$ | $82.5 \pm 4.5$ | $79.2 \pm 4.4$ | $33.4 \pm 5.7$ |
| pow | $\frac{1}{2}\mu$ | $68.8 \pm 10.6$ | $70.1 \pm 10.4$ | $\mathbf{83.2 \pm 9.3}$ | $6.2 \pm 6.7$ | $6.2 \pm 6.7$ | $6.2 \pm 6.7$ | $6.2 \pm 6.7$ |
| | $1\mu$ | $68.8 \pm 10.6$ | $\underline{72.7 \pm 9.9}$ | $\mathbf{84.2 \pm 9.2}$ | $6.2 \pm 6.7$ | $6.2 \pm 6.7$ | $6.2 \pm 6.7$ | $6.2 \pm 6.7$ |
| | $2\mu$ | $\underline{78.6 \pm 9.0}$ | $75.1 \pm 9.5$ | $\mathbf{84.1 \pm 9.2}$ | $6.2 \pm 6.7$ | $6.2 \pm 6.7$ | $17.5 \pm 12.0$ | $6.2 \pm 6.7$ |
| nets | $\frac{1}{2}\mu$ | $36.6 \pm 10.5$ | $36.6 \pm 10.5$ | $\mathbf{58.1 \pm 8.8}$ | $1.8 \pm 2.2$ | $1.8 \pm 2.2$ | $1.8 \pm 2.2$ | $1.8 \pm 2.2$ |
| | $1\mu$ | $\underline{36.6 \pm 10.5}$ | $36.6 \pm 10.4$ | $\mathbf{58.9 \pm 8.7}$ | $1.8 \pm 2.2$ | $1.8 \pm 2.2$ | $15.0 \pm 8.7$ | $1.8 \pm 2.2$ |
| | $2\mu$ | $\underline{40.3 \pm 10.4}$ | $36.6 \pm 10.4$ | $\mathbf{58.1 \pm 8.8}$ | $1.8 \pm 2.2$ | $5.3 \pm 5.3$ | $22.3 \pm 9.7$ | $1.8 \pm 2.2$ |

Table 3: F1 Score (F1, %) $\pm$ error of SR and ONMI in the asymmetric setting (training: angel, testing: demon), $\tau = 0.5$. Best results are in bold, second best are underlined

| | | Asymmetric setting — training: angel; testing: demon | | | | | | |
|---|---|---|---|---|---|---|---|---|
| Dataset | $\beta$ | ODRL(0.5$\mu$) | ODRL(1.0$\mu$) | ODRL(2.0$\mu$) | Random | Degree | Betweenness | Roam |
| kar | $\frac{1}{2}\mu$ | $57.4 \pm 14.5$ | $\mathbf{73.0 \pm 12.3}$ | $69.2 \pm 11.3$ | $25.8 \pm 13.6$ | $65.0 \pm 9.2$ | $61.9 \pm 10.0$ | $19.6 \pm 14.0$ |
| | $1\mu$ | $76.5 \pm 12.3$ | $76.2 \pm 11.9$ | $\mathbf{78.8 \pm 12.4}$ | $28.8 \pm 14.9$ | $69.1 \pm 7.9$ | $68.8 \pm 7.8$ | $29.3 \pm 15.4$ |
| | $2\mu$ | $\underline{87.5 \pm 11.4}$ | $\mathbf{88.0 \pm 11.0}$ | $81.7 \pm 11.2$ | $31.2 \pm 14.4$ | $76.5 \pm 5.8$ | $73.6 \pm 6.1$ | $30.4 \pm 13.4$ |
| words | $\frac{1}{2}\mu$ | $27.5 \pm 6.5$ | $45.7 \pm 5.8$ | $31.0 \pm 6.7$ | $2.7 \pm 2.8$ | $\underline{49.1 \pm 5.5}$ | $\mathbf{50.9 \pm 5.1}$ | $9.2 \pm 4.6$ |
| | $1\mu$ | $32.7 \pm 6.6$ | $50.4 \pm 5.5$ | $35.4 \pm 6.6$ | $6.1 \pm 4.4$ | $\underline{60.5 \pm 4.3}$ | $\mathbf{61.9 \pm 4.1}$ | $9.1 \pm 4.6$ |
| | $2\mu$ | $44.6 \pm 6.0$ | $54.3 \pm 5.5$ | $43.5 \pm 6.2$ | $9.3 \pm 4.3$ | $\underline{65.6 \pm 3.5}$ | $\mathbf{69.0 \pm 3.2}$ | $14.6 \pm 5.4$ |
| vote | $\frac{1}{2}\mu$ | $50.7 \pm 4.9$ | $\mathbf{54.2 \pm 4.9}$ | $\underline{51.8 \pm 4.5}$ | $8.1 \pm 3.8$ | $16.8 \pm 4.8$ | $16.4 \pm 4.9$ | $7.0 \pm 3.6$ |
| | $1\mu$ | $51.4 \pm 4.8$ | $\mathbf{64.0 \pm 4.5}$ | $\underline{52.6 \pm 4.5}$ | $8.1 \pm 3.8$ | $60.5 \pm 4.6$ | $51.1 \pm 4.8$ | $9.6 \pm 3.8$ |
| | $2\mu$ | $57.5 \pm 4.6$ | $61.9 \pm 4.6$ | $56.1 \pm 4.4$ | $8.1 \pm 3.8$ | $\mathbf{78.7 \pm 2.7}$ | $\underline{70.5 \pm 3.8}$ | $12.5 \pm 4.6$ |
| pow | $\frac{1}{2}\mu$ | $\mathbf{92.6 \pm 5.6}$ | $89.3 \pm 6.2$ | $\underline{89.6 \pm 6.2}$ | $0.0 \pm 0.0$ | $8.4 \pm 8.1$ | $0.0 \pm 0.0$ | $0.0 \pm 0.0$ |
| | $1\mu$ | $\mathbf{92.0 \pm 5.7}$ | $\underline{90.6 \pm 6.0}$ | $90.3 \pm 6.1$ | $0.0 \pm 0.0$ | $8.4 \pm 8.1$ | $0.0 \pm 0.0$ | $0.0 \pm 0.0$ |
| | $2\mu$ | $\underline{91.9 \pm 5.8}$ | $\mathbf{92.0 \pm 5.7}$ | $90.2 \pm 6.1$ | $0.0 \pm 0.0$ | $8.4 \pm 8.1$ | $2.9 \pm 4.2$ | $0.0 \pm 0.0$ |
| nets | $\frac{1}{2}\mu$ | $52.9 \pm 9.2$ | $\mathbf{64.2 \pm 8.2}$ | $52.9 \pm 9.1$ | $0.0 \pm 0.0$ | $4.8 \pm 5.5$ | $0.0 \pm 0.0$ | $0.0 \pm 0.0$ |
| | $1\mu$ | $\underline{54.7 \pm 9.0}$ | $\mathbf{64.2 \pm 8.2}$ | $52.9 \pm 9.2$ | $1.6 \pm 2.4$ | $10.9 \pm 6.6$ | $12.4 \pm 7.9$ | $0.0 \pm 0.0$ |
| | $2\mu$ | $\underline{54.7 \pm 9.0}$ | $\mathbf{64.2 \pm 8.2}$ | $54.7 \pm 9.0$ | $1.6 \pm 2.4$ | $18.0 \pm 8.8$ | $13.8 \pm 7.8$ | $0.0 \pm 0.0$ |

A first bottleneck lies in scalability: although our method is inherently parallelizable, training requires repeated calls to the community detection oracle, which is computationally expensive and

often lacks efficient GPU implementations. This makes training on massive graphs with billions of nodes prohibitively slow. Future work could explore the use of surrogate models to approximate the community detection function, providing faster, even though less precise, feedback to the agent.

Second, our current formulation assumes static graphs, whereas real-world networks are dynamic. Policies trained on a single snapshot may quickly become obsolete. Extending the framework to dynamic or temporal graphs, possibly through continual learning, would improve applicability.

Third, our work considers a single target node. In practice, multiple users may wish to hide their memberships simultaneously, which calls for multi-agent formulations that can model cooperation or competition among agents.

Beyond these assumptions, future research should also explore alternative definitions of the hiding objective, extend evaluations to larger graphs and different community detection algorithms, consider broader hyperparameter settings, and assess unintended side effects of edge modifications, such as their impact on the community assignments of non-target nodes.

## 8 CONCLUSION

In this paper, we tackled the problem of community membership hiding (CMH) under the realistic and challenging setting of overlapping communities. We further showed that a simple injection strategy is sufficient to solve the non-overlapping case, highlighting the added complexity of the overlapping scenario. To the best of our knowledge, this is the first formalization of CMH in the overlapping setting and the first principled solution to it. We proposed a deep reinforcement learning framework that strategically injects proxy nodes and performs edge modifications to effectively obscure a target node's affiliation from its original community. Our empirical results across multiple real-world datasets show that the method consistently outperforms strong baselines in both effectiveness and efficiency. Overall, our work establishes a foundation for privacy-preserving graph analysis, with promising directions in scaling to dynamic and large-scale networks.

## REPRODUCIBILITY

We have made every effort to ensure the reproducibility of our work. The source code for our ODRL agent, the experimental setup, and all baseline implementations is available as supplementary material at `https://anonymous.4open.science/r/overlapping_comm_detect-EC02/README.md`. The datasets used in our experiments are standard, publicly available benchmarks. Section 6 and Table 1 provide detailed descriptions and references to their sources. The specifics of our proposed methodology, including the Markov decision process (MDP) formulation, network architectures, and training procedure, are detailed in Sections 5 and 6. Further implementation details and key hyperparameters are provided in Appendix B to facilitate the full replication of our results.

## ETHICAL CONSIDERATIONS

This work contributes to the field of privacy-preserving machine learning, but we recognize its dual-use nature. On the one hand, the developed method serves as a powerful tool for safeguarding user privacy. It can empower individuals, such as journalists, activists, or members of vulnerable groups, to protect themselves from algorithmic profiling and unwanted exposure by controlling their visibility in online social networks. This aligns with the principles of data autonomy and the "right to be forgotten".

On the other hand, like any privacy-enhancing technology, this method could be misused by malicious actors. For instance, it could be employed to conceal criminal or coordinated inauthentic behavior, thereby evading detection by graph-based security and content moderation systems. Furthermore, modifications made to hide one user may have unintended side effects on the community assignments of their neighbors.

We believe the potential benefits for individual privacy are significant. We advocate for the responsible deployment of such technologies. If implemented by platform providers, this capability should

be accompanied by robust safeguards, such as access controls, monitoring for anomalous activity, and transparent policies to mitigate the risk of misuse.

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

## GenAI Disclosure Statement

We used GPT-5 to identify and correct grammatical errors and typos, and to improve overall writing quality. No AI tools were used at any other stage of this work to ensure complete academic integrity.

## A  Community Detection Algorithms

To ensure the robustness of our evaluation, we test our evasion strategy against algorithms that rely on fundamentally different definitions of graph clustering.

### A.1  Non-Overlapping Algorithms

These methods enforce a strict partition where each node $u$ belongs to exactly one community ($|f(u, G)| = 1$). Below, we discuss in detail the detection algorithms that are used in this work.

- **Greedy Modularity**: Proposed by Brandes et al. (2008), it is a hierarchical agglomerative method. It initializes every node as its own community and iteratively merges the pair of communities that yields the maximum increase in global modularity ($\Delta Q$). The process stops when no merge can further increase modularity. In many practical sparse networks, it behaves closer to $O(m \log n)$.

- **Louvain**: A heuristic method by Blondel et al. (2008) that optimizes modularity in two repeating phases. First, it iterates through nodes, moving each node $u$ to the neighbor's community that maximizes the local gain in modularity. Second, it aggregates the graph, treating the formed communities as single super-nodes. This allows the algorithm to detect hierarchical structures and scales efficiently to large networks. It has been experimentally proven to run in $O(n \log n)$ time.

- **Walktrap**: Developed by Pons & Latapy (2005), this algorithm utilizes random walks. The core intuition is that random walks are likely to get "trapped" within densely connected subgraphs (communities). It defines a distance metric between nodes based on the transition probabilities of random walks of length $t$ and uses this metric to perform hierarchical clustering. In many practical sparse networks, it behaves closer to $O(n^2)$.

### A.2  Overlapping algorithms

These methods allow for cover solutions where nodes may belong to multiple communities or none ($|f(u, G)| \geq 0$). Below, we discuss in detail the detection algorithms that are used in this work.

- **Demon**: The Democratic Estimate of the Modular Organization of a Network (Coscia et al., 2012) is a local-first approach. For every node $u$, it extracts the ego-network $E(u)$ and removes $u$ itself. It then applies a simple Label Propagation algorithm to the remaining neighbors to identify local communities. Finally, the algorithm collects all local communities found across all ego-networks and merges those that share a significant portion of nodes. This merging process is controlled by a threshold $\phi$; if the similarity between two communities exceeds $\phi$, they are combined to avoid redundant clusters.

- **Angel**: Rossetti (2020) proposes a node-centric approach designed for efficiency. The algorithm operates by incrementally assembling communities around "core" nodes, gradually expanding them by adding neighbors that maintain high structural density. Crucially, to handle the vast number of potential overlapping structures, `angel` employs a merging threshold $\phi$. After identifying candidate communities, the algorithm computes their similarity; if the overlap between any two communities exceeds $\phi$, they are merged into a single community. This mechanism ensures that

the algorithm returns distinct, meaningful groups rather than highly redundant variations of the same dense subgraph.

## B  IMPLEMENTATION DETAILS

**Network Architecture**   The agent's policy and value functions are parameterized by a neural network composed of three main components: a shared graph encoder, an actor head, and a critic head, all built upon a shared state representation.

**Shared Graph Encoder.** The encoder maps input node features $x \in \mathbb{R}^{|V| \times d_{in}}$ to rich node embeddings $h \in \mathbb{R}^{|V| \times d_h}$. It consists of four GCN layers (Kipf & Welling, 2016), each employing PairNorm normalization (Zhao & Akoglu, 2019) and ELU nonlinearities (Clevert et al., 2015). To create multi-scale representations and mitigate oversmoothing, the outputs of each layer (each with dimensionality $d_h/4$) are concatenated, in a style similar to jumping knowledge networks (Xu et al., 2018).

**State Representation.** From the node embeddings produced by the encoder, we construct an aggregated state context. Specifically, we compute representations for the target node, the set of proxy nodes, and the entire graph:

$$h_{\text{target}} = h_{i_u}, \qquad h_{\text{proxy}} = \frac{1}{|P|} \sum_{p \in P} h_{i_p}, \qquad h_{\text{global}} = \frac{1}{|V|} \sum_{v \in V} h_{i_v} \qquad (8)$$

These vectors are concatenated and passed through a linear projection with LayerNorm and an ELU activation. To capture temporal dependencies within an episode, this resulting vector is processed by a single-layer GRU (Cho et al., 2014). The GRU's hidden state serves as the final shared representation, $h_{\text{shared}}$, for the current step.

**Actor and Critic Heads.** From this shared backbone, the network branches into actor and critic branches. The **actor** implements the factored policy. First, a linear projection maps $h_{\text{shared}}$ to logits over the $|P|+1$ candidate nodes (the target and its proxies). Then, for each candidate, an independent linear head takes the concatenation of its node embedding and the shared state, $[h_{\text{shared}}; h_{\text{actor}}]$, and produces logits over the $2(|V|-1)$ available actions. The **critic** is a two-layer MLP with ELU nonlinearities that maps the shared representation $h_{\text{shared}}$ directly to a scalar state-value estimate, $V_\theta(\cdot)$.

**Training Details**   We train the ODRL agent for $5,000$ episodes using the RMSProp optimizer (Hinton, 2012) with a learning rate of $\eta = 5 \times 10^{-4}$. Our PPO implementation uses a clipping parameter $\epsilon = 0.1$ and Generalized Advantage Estimation (GAE) (Schulman et al., 2016) with $\lambda = 0.95$ and $\gamma = 0.99$, performing four updates per episode. For the loss function in Eq. (5), the coefficients are set to $c_v = 1.0$ and $c_{clip} = 0.1$, while the entropy coefficient $c_{ent}$ is linearly annealed from $10^{-2}$ to $10^{-4}$ to balance exploration and exploitation. The hidden dimension for all actor and critic networks is $d_h = 32$. To promote training diversity, we resample the target node every 5 episodes and the target community every 50 episodes. The computational cost associated with our training procedure, measured in wall-clock time (minutes) is available in Table 5.

**Sensitivity Analysis of the Merging Threshold**   We analyze the impact of the merging threshold $\phi$ on community detection granularity (Table 4). At $\phi = 0.5$, `angel` significantly under-segments networks, detecting only 6 communities in `vote` versus 37 by `demon`. Setting $\phi = 0.8$ resolves this discrepancy, producing comparable community counts across algorithms that scale appropriately with network size. We therefore adopt $\phi = 0.8$ as a merging threshold for all evasion experiments.

## C  ADDITIONAL RESULTS

In this section, we provide additional experimental results to supplement the analysis presented in the main paper. While the main text focuses on the F1 score to provide a balanced view of performance, here we present the disaggregated Success Rate (SR) metric. Specifically, Table 6 details the SR for the symmetric setting, while Table 7 shows the SR for the asymmetric setting. These tables offer a more granular perspective on the effectiveness of our method in achieving the primary hiding

Table 4: Number of communities detected by `demon` and `angel` across varying $\phi$ thresholds

| Graph Properties | | | demon ($\phi$) | | | | angel ($\phi$) | | | |
|---|---|---|---|---|---|---|---|---|---|---|
| Dataset | $|V|$ | $|E|$ | 0.5 | 0.6 | 0.7 | 0.8 | 0.5 | 0.6 | 0.7 | 0.8 |
| kar | 34 | 78 | 3 | 4 | 5 | 5 | 3 | 3 | 6 | 2 |
| vote | 889 | 2,914 | 37 | 57 | 107 | 188 | 6 | 15 | 81 | 161 |
| nets | 1,589 | 2,742 | 148 | 152 | 153 | 159 | 141 | 145 | 327 | 123 |

Table 5: Wall-clock time (in minutes) for the complete experimental pipeline across each dataset.

| Dataset | Time (min) |
|---|---|
| kar | 113.88 |
| words | 364.08 |
| vote | 504.30 |
| pow | 258.72 |
| nets | 195.00 |

objective across various datasets and budgets. Figures 5 and 6 show the F1 score performance with budgets $\beta \in \{\frac{1}{2}\mu, 2\mu\}$.

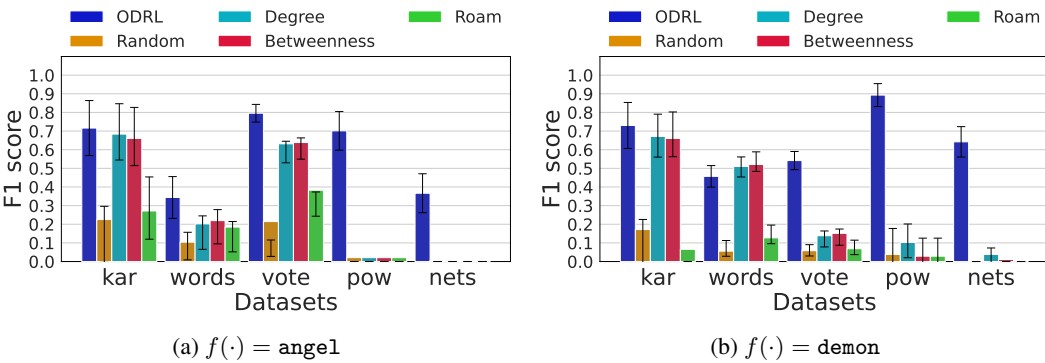

(a) $f(\cdot) = $ `angel`  (b) $f(\cdot) = $ `demon`

Figure 5: F1 scores of SR and ONMI for the *ODRL Agent* and baselines in the symmetric (Figure 5a) and asymmetric (Figure 5b) scenarios, with parameters $k = \mu$, $p = 0.5$, $\beta = \frac{1}{2}\mu$.

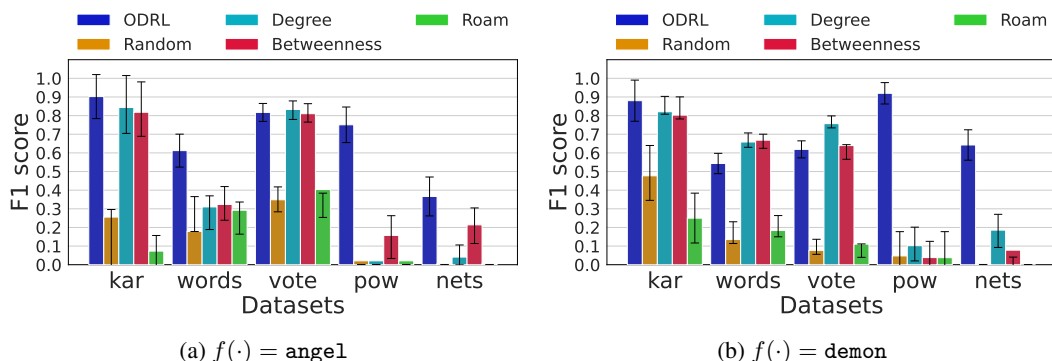

(a) $f(\cdot) = \texttt{angel}$      (b) $f(\cdot) = \texttt{demon}$

Figure 6: F1 scores of SR and ONMI for the *ODRL Agent* and baselines in the symmetric (Figure 6a) and asymmetric (Figure 6b) scenarios, with parameters $k = \mu$, $p = 0.5$, $\beta = 2\mu$.

Table 6: Success Rate (SR, %) $\pm$ error of SR in the symmetric setting (`angel`), $\tau = 0.5$.

| Dataset | $\beta$ | ODRL($0.5\mu$) | ODRL($1.0\mu$) | ODRL($2.0\mu$) | Random | Degree | Betweenness | Roam |
|---|---|---|---|---|---|---|---|---|
| | | | | **Symmetric setting** — training: angel; testing: angel | | | | |
| kar | $\frac{1}{2}\mu$ | $48.0 \pm 19.6$ | $64.0 \pm 18.8$ | $44.0 \pm 19.5$ | $40.0 \pm 19.2$ | $60.0 \pm 19.2$ | $56.0 \pm 19.5$ | $16.0 \pm 14.4$ |
| | $1\mu$ | $76.0 \pm 16.7$ | $76.0 \pm 16.7$ | $76.0 \pm 16.7$ | $40.0 \pm 19.2$ | $76.0 \pm 16.7$ | $76.0 \pm 16.7$ | $4.0 \pm 5.8$ |
| | $2\mu$ | $100.0 \pm 0.0$ | $100.0 \pm 0.0$ | $100.0 \pm 0.0$ | $52.0 \pm 19.6$ | $100.0 \pm 0.0$ | $88.0 \pm 12.4$ | $4.0 \pm 5.8$ |
| words | $\frac{1}{2}\mu$ | $18.0 \pm 8.0$ | $21.1 \pm 8.4$ | $21.3 \pm 8.5$ | $9.5 \pm 5.9$ | $16.8 \pm 7.5$ | $18.9 \pm 7.9$ | $17.9 \pm 7.7$ |
| | $1\mu$ | $25.8 \pm 9.1$ | $30.0 \pm 9.5$ | $29.2 \pm 9.4$ | $10.5 \pm 6.2$ | $21.1 \pm 8.2$ | $21.1 \pm 8.2$ | $21.1 \pm 8.2$ |
| | $2\mu$ | $37.1 \pm 10.0$ | $62.2 \pm 10.0$ | $38.2 \pm 10.1$ | $10.5 \pm 6.2$ | $25.3 \pm 8.7$ | $26.3 \pm 8.9$ | $29.5 \pm 9.2$ |
| vote | $\frac{1}{2}\mu$ | $84.8 \pm 3.8$ | $79.6 \pm 4.8$ | $94.5 \pm 2.5$ | $11.7 \pm 3.5$ | $50.5 \pm 5.4$ | $51.1 \pm 5.4$ | $17.8 \pm 4.2$ |
| | $1\mu$ | $89.3 \pm 3.3$ | $83.6 \pm 4.4$ | $95.5 \pm 2.3$ | $13.2 \pm 3.7$ | $75.7 \pm 4.7$ | $63.7 \pm 5.2$ | $17.8 \pm 4.2$ |
| | $2\mu$ | $92.6 \pm 2.8$ | $84.7 \pm 4.3$ | $95.8 \pm 2.2$ | $13.5 \pm 3.7$ | $87.1 \pm 3.6$ | $79.7 \pm 4.4$ | $20.9 \pm 4.4$ |
| pow | $\frac{1}{2}\mu$ | $54.1 \pm 12.5$ | $55.7 \pm 12.5$ | $80.4 \pm 10.4$ | $3.2 \pm 3.8$ | $3.2 \pm 3.8$ | $3.2 \pm 3.8$ | $3.2 \pm 3.8$ |
| | $1\mu$ | $54.1 \pm 12.5$ | $59.0 \pm 12.3$ | $82.1 \pm 10.0$ | $3.2 \pm 3.8$ | $3.2 \pm 3.8$ | $3.2 \pm 3.8$ | $3.2 \pm 3.8$ |
| | $2\mu$ | $67.2 \pm 11.8$ | $62.3 \pm 12.2$ | $82.1 \pm 10.0$ | $3.2 \pm 3.8$ | $3.2 \pm 3.8$ | $9.7 \pm 7.4$ | $3.2 \pm 3.8$ |
| nets | $\frac{1}{2}\mu$ | $22.6 \pm 8.0$ | $22.6 \pm 8.0$ | $41.5 \pm 8.9$ | $0.9 \pm 1.3$ | $0.9 \pm 1.3$ | $0.9 \pm 1.3$ | $0.9 \pm 1.3$ |
| | $1\mu$ | $22.6 \pm 8.0$ | $22.6 \pm 8.0$ | $42.4 \pm 8.9$ | $0.9 \pm 1.3$ | $0.9 \pm 1.3$ | $8.1 \pm 5.1$ | $0.9 \pm 1.3$ |
| | $2\mu$ | $25.5 \pm 8.3$ | $22.6 \pm 8.0$ | $41.5 \pm 8.9$ | $0.9 \pm 1.3$ | $2.7 \pm 2.9$ | $12.6 \pm 6.2$ | $0.9 \pm 1.3$ |

Table 7: Success Rate (SR, %) $\pm$ error of SR in the symmetric setting (training: `angel`, testing: `demon`), $\tau = 0.5$.

| Dataset | $\beta$ | ODRL($0.5\mu$) | ODRL($1.0\mu$) | ODRL($2.0\mu$) | Random | Degree | Betweenness | Roam |
|---|---|---|---|---|---|---|---|---|
| | | | | **Asymmetric setting** — training: angel; testing: demon | | | | |
| kar | $\frac{1}{2}\mu$ | $44.4 \pm 16.2$ | $63.9 \pm 15.7$ | $63.3 \pm 13.5$ | $15.6 \pm 10.6$ | $57.8 \pm 14.4$ | $53.3 \pm 14.6$ | $11.1 \pm 9.2$ |
| | $1\mu$ | $72.2 \pm 14.6$ | $69.4 \pm 15.0$ | $87.8 \pm 9.2$ | $17.8 \pm 11.2$ | $66.7 \pm 13.8$ | $66.7 \pm 13.8$ | $17.8 \pm 11.2$ |
| | $2\mu$ | $94.4 \pm 6.5$ | $94.4 \pm 6.5$ | $95.9 \pm 4.8$ | $20.0 \pm 11.7$ | $82.2 \pm 11.2$ | $77.8 \pm 12.1$ | $20.0 \pm 11.7$ |
| words | $\frac{1}{2}\mu$ | $17.1 \pm 5.0$ | $34.2 \pm 6.1$ | $19.6 \pm 5.3$ | $1.4 \pm 1.4$ | $36.9 \pm 6.3$ | $38.7 \pm 6.4$ | $5.0 \pm 2.9$ |
| | $1\mu$ | $21.2 \pm 5.4$ | $40.6 \pm 6.3$ | $23.4 \pm 5.7$ | $3.2 \pm 2.3$ | $52.7 \pm 6.6$ | $55.4 \pm 6.5$ | $5.0 \pm 2.9$ |
| | $2\mu$ | $33.2 \pm 6.3$ | $44.4 \pm 6.4$ | $31.3 \pm 6.2$ | $5.0 \pm 2.9$ | $63.1 \pm 6.3$ | $67.6 \pm 6.2$ | $8.1 \pm 3.6$ |
| vote | $\frac{1}{2}\mu$ | $36.3 \pm 4.8$ | $39.7 \pm 5.0$ | $37.5 \pm 4.6$ | $4.2 \pm 2.1$ | $9.3 \pm 3.0$ | $9.0 \pm 3.0$ | $3.7 \pm 2.0$ |
| | $1\mu$ | $37.1 \pm 4.9$ | $51.5 \pm 5.1$ | $38.4 \pm 4.6$ | $4.2 \pm 2.1$ | $46.2 \pm 5.2$ | $36.1 \pm 5.0$ | $5.1 \pm 2.3$ |
| | $2\mu$ | $43.7 \pm 5.0$ | $48.5 \pm 5.1$ | $41.9 \pm 4.7$ | $4.2 \pm 2.1$ | $71.8 \pm 4.7$ | $60.0 \pm 5.1$ | $6.8 \pm 2.6$ |
| pow | $\frac{1}{2}\mu$ | $88.8 \pm 6.9$ | $82.5 \pm 8.3$ | $83.8 \pm 8.1$ | $0.0 \pm 0.0$ | $4.4 \pm 4.6$ | $0.0 \pm 0.0$ | $0.0 \pm 0.0$ |
| | $1\mu$ | $87.5 \pm 7.2$ | $85.0 \pm 7.8$ | $85.0 \pm 7.8$ | $0.0 \pm 0.0$ | $4.4 \pm 4.6$ | $0.0 \pm 0.0$ | $0.0 \pm 0.0$ |
| | $2\mu$ | $87.5 \pm 7.2$ | $87.5 \pm 7.2$ | $85.0 \pm 7.8$ | $0.0 \pm 0.0$ | $4.4 \pm 4.6$ | $1.5 \pm 2.2$ | $0.0 \pm 0.0$ |
| nets | $\frac{1}{2}\mu$ | $36.4 \pm 8.6$ | $47.9 \pm 8.9$ | $36.4 \pm 8.6$ | $0.0 \pm 0.0$ | $2.5 \pm 2.6$ | $0.0 \pm 0.0$ | $0.0 \pm 0.0$ |
| | $1\mu$ | $38.0 \pm 8.6$ | $47.9 \pm 8.9$ | $36.4 \pm 8.6$ | $0.8 \pm 1.2$ | $5.8 \pm 4.2$ | $6.6 \pm 4.4$ | $0.0 \pm 0.0$ |
| | $2\mu$ | $38.0 \pm 8.6$ | $47.9 \pm 8.9$ | $38.0 \pm 8.6$ | $0.8 \pm 1.2$ | $9.9 \pm 5.3$ | $7.4 \pm 4.7$ | $0.0 \pm 0.0$ |

