# OpenReview forum: "Evading Overlapping Community Detection via Proxy Node Injection"
_ICLR.cc/2026/Conference — Submitted to ICLR 2026_

### Official Review · Reviewer_Ji1w · 2025-10-29

**Soundness:** 2
**Presentation:** 2
**Contribution:** 2
**Rating:** 4
**Confidence:** 5

**Summary:**

This paper addresses the Community Membership Hiding problem in the challenging, realistic setting of overlapping communities. The authors propose a deep reinforcement learning framework that injects proxy nodes and learns an optimal edge modification policy to hide a target node's affiliation. Experiments show this method significantly outperforms baselines.

**Strengths:**

1. The paper addresses the significant problem of privacy in graph analysis by formally defining and tackling Community Membership Hiding in the realistic setting of overlapping communities.

2. The proposed solution combines a deep reinforcement learning framework with a proxy node injection technique to learn effective graph modifications.

3. The paper provides an empirical evaluation on real-world datasets, demonstrating that the proposed method significantly outperforms baselines.

**Weaknesses:**

1. The paper's claim to be the "first to formalize and address CMH under overlapping community detection" may need to be revisited. The problem of hiding membership in overlapping communities has been explored in prior works [1][2][3]. The authors should refine their contribution statement to more clearly distinguish their work from these existing studies and conduct a more comprehensive survey of related literature.

2. The current baselines are primarily variations applied within the proposed proxy-injection framework. The comparison should include established, state-of-the-art methods specifically designed for overlapping community hiding.

3. The proposed ODRL method requires repeated queries to a community detection oracle, which can be computationally intensive. The current experiments are conducted on relatively small graphs. To demonstrate practical viability for real-world applications, it would be beneficial to evaluate the method on larger-scale datasets and discuss potential optimization strategies to mitigate the computational overhead.

4. The paper could be further strengthened by including theoretical analysis to provide a more formal justification for the proposed ODRL framework's design and efficacy.

References:

[1] Liu D, Yang G, Wang Y, et al. How to protect ourselves from overlapping community detection in social networks[J]. IEEE Transactions on Big Data, 2022, 8(4): 894-904.

[2] Yang G, Wang Y, Chang Z, et al. Overlapping community hiding method based on multi-level neighborhood information[J]. Symmetry, 2022, 14(11): 2328.

[3] Han Z, Shi L, Wang Y, et al. CoHide: Overlapping community hiding algorithm based on multi-criteria learning optimization[J]. Peer-to-Peer Networking and Applications, 2025, 18(2): 94.

**Questions:**

1. Could the authors clarify the specific novelty of their problem formulation or technical approach that distinguishes it from prior work which also appears to address overlapping community hiding?

2. To better contextualize the method's performance, could the authors elaborate on why established, external baselines from the literature were not included?

3. Given that the proposed framework requires repeated calls to a community detection oracle, could the authors provide the practical training and inference runtimes for the experiments? Furthermore, could the authors discuss the optimization strategies for scaling this approach to much larger, real-world graphs?

---

> ### Author Response · Authors · 2025-11-20
> **Response to Reviewer Ji1w**
>
> We thank the Reviewer for the constructive feedback. We address each point below and hope these clarifications resolve remaining concerns.
>
> **Q1./W1. Novelty relative to prior overlapping-hiding works**
>
> We appreciate the reviewer pointing out these related works. While they share the broad domain of "overlapping community hiding," we respectfully submit that our work addresses a fundamentally different problem formulation than the ones cited ([1] Liu et al., 2022; [2] Yang et al., 2022; [3] Han et al., 2025).
> The key distinctions lie in the hiding objective and the metric for success: prior works typically define the "overlapping area" itself as the danger zone.
> - [1] and [3] explicitly aim to remove the target node from the set of overlapping nodes entirely (i.e., forcing the node into a single community).
> - [2] (MLNI) aim to move a node out of a specific community, but their formulation relies on maximizing the difference in discrete "membership probabilities" derived from neighborhood layers.
>
> In contrast, our work formalizes the problem as hiding a specific sensitive affiliation, regardless of whether the node remains in an overlapping area or joins new communities. We argue that in realistic social graphs, being in an "overlapping area" is not a concerning situation, whereas membership in a specific community is.
>
> A fundamental limitation of prior formulations ([1], [2], [3]) is their reliance on discrete metrics, such as community counts or binary membership flags. These measures fail to capture the dynamic nature of community detection; for instance, if an attack prompts the detector to generate a 'new' community ID that retains a nearly identical member composition, discrete methods would erroneously classify this as a successful hiding.
> In contrast, our solution employs set-similarity metrics to compare the original community with post-perturbation structures rigorously. This prevents false positives caused by trivial label changes, ensuring that 'success' in our framework implies the genuine removal of the privacy leak. We will include this discussion in the related work section if the paper is accepted.
>
>
> **Q2./W2. External baselines**
>
> As detailed above, the baselines [1-3] address a fundamentally different optimization problem (Overlapping Area Avoidance) and utilize discrete objective functions, which prevent a direct, fair comparison with our continuous, similarity-based formulation. However, we observe that the strategies employed in these works rely heavily on standard topological heuristics. Consequently, the baselines we independently selected, Random, Degree, and Betweenness, align directly with the core methods reported in the literature:
> - [1] employ Random, Degree, and Betweenness Hiding.
> - [2] and [3] similarly rely on Random and Degree-based strategies
>
> Thus, by evaluating ODRL against these standard heuristics, we are effectively benchmarking against the algorithmic core of prior work, albeit applied within our novel proxy-injection setting, a mechanism not explored by [1-3].
> Direct reproduction is also infeasible due to the absence of publicly available implementations.
>
> **Q3./W3. Practical runtimes and scalability**
>
> We acknowledge the computational cost of the oracle. We report the wall-clock training times from Table 2 (See **W2** in the comment to reviewer `up1n`).
>
> As a possible optimization strategy, two approaches are promising:
> - *Surrogate Modeling*: Training a GNN to approximate the community detection outputs, replacing slow oracle calls with efficient forward passes.
> - *Local Community Detection*: Instead of running global detection (which is prohibitive for large graphs), we can use local community detection algorithms that only expand around the target node. This reduces the complexity from graph-size-dependent to community-size-dependent.
>
> **W4. Theoretical justification for ODRL**
>
> ODRL is motivated by three structural aspects of the problem:
> - *Combinatorial Complexity*: Graph-modification problems under constraints are generally intractable; even centrality disguise (a more straightforward objective) is NP-hard (Waniek et al., 2018). Community-membership hiding involves mesoscale structure with non-local dependencies, making exact optimization unattainable. RL offers a principled way to search such large discrete spaces.
> - *Limitations of Heuristics in Overlapping Settings*: As shown in Section 4.3 and Figure 3, heuristics that work reasonably well for non-overlapping communities degrade sharply when nodes can sustain multiple memberships. Effective hiding often requires multi-step, non-greedy perturbations, which RL naturally learns.
> - *Non-Differentiable Objectives*: Community detection is a black-box, non-differentiable oracle, precluding gradient-based methods. RL is explicitly designed to optimize such reward structures. This rationale aligns with insights from Bernini et al. (2024) in the non-overlapping variant of the problem.

---

> ### Author Response · Authors · 2025-11-27
>
> Dear Reviewer Ji1w,
>
> Thank you very much again for your time and effort in reviewing our paper. To summarize our previous response and follow up respectfully, we would like to highlight the concrete steps we have taken to address your valuable comments:
>
> * **Novelty (Q1/W1)**: We distinguished our goal (Specific Affiliation Hiding) from prior works that target Overlapping Area Avoidance (forcing nodes into single communities). Our continuous similarity metrics avoid the false positives inherent in their discrete measures. We included this comparison in the Related Works section.
>
> * **Baselines (Q2/W2)**: While direct comparison is incompatible due to differing objectives, our evaluation against standard heuristics (Random/Degree/Betweenness) covers the algorithmic core of the cited works.
>
> * **Scalability & Theory (Q3-W4)**: We provided wall-clock training times to ensure transparency. We further justified ODRL as necessary for this combinatorial, black-box problem, where greedy heuristics, effective in non-overlapping settings, fail.
>
> We value your feedback and remain available to address any further questions you may have.

---

### Official Review · Reviewer_up1n · 2025-10-31

**Soundness:** 2
**Presentation:** 3
**Contribution:** 2
**Rating:** 4
**Confidence:** 4

**Summary:**

This paper formalizes community membership hiding (CMH) for overlapping community detection, where nodes naturally belong to multiple communities. The authors propose ODRL, a deep reinforcement learning agent employing Proximal Policy Optimization (PPO) with a factored policy to inject proxy nodes and modify edges, concealing a target node's community affiliation. Experiments on five real-world datasets demonstrate substantial F1 score improvements over heuristic baselines in both symmetric (same detector for training/testing) and asymmetric (different detectors) settings. While the problem formulation is novel and results promising, concerns regarding experimental rigor, scalability, detectability analysis, and practical deployment warrant careful consideration.

**Strengths:**

- Section 4 The formalization of CMH for overlapping detection is a meaningful contribution. Figures 2-3 compellingly show that naïve proxy injection achieves F1 scores of 0.6-0.9 for non-overlapping detectors but drops below 0.3 for overlapping detectors, clearly establishing problem non-triviality.​
- Section 5: The factored policy decomposition reduces computational complexity from O(|V|²) to O(|V|) while enabling action masking. The 4-layer GCN with jumping knowledge connections and GRU temporal representation reflects careful design. The reward function (Eq. 3) balances immediate success with incremental progress via similarity differential δtsim​.
- Tables 2-3, Figures 4: The asymmetric setting (training on ANGEL, testing on DEMON) demonstrates cross-algorithm generalization crucial for real-world deployment. ODRL maintains strong performance with 20-70 percentage point F1 improvements over baselines on large datasets.​
- ODRL's advantage increases with graph size and complexity, achieving 60-90% F1 scores on pow and nets where baselines approach zero, suggesting meaningful structural pattern learning.​

**Weaknesses:**

- Baselines use fixed k=μ while ODRL explores k∈{0.5μ, μ, 2μ}, making it unclear whether gains stem from the learned policy or expanded budget exploration. Missing are comparisons with Silvestri et al. (2025)'s gradient-based optimization and sophisticated node injection attacks (GANI, TDGIA, G-NIA) from adversarial graph learning. Critical ablations are absent, no evaluation of ODRL without proxies (P=∅) or flat vs. factored policy comparisons.​​
- While acknowledging expensive oracle calls, no wall-clock times, oracle queries per episode, or memory requirements are reported. Excluding fb-75 because experiments "could not be completed within a reasonable timeframe" is concerning for modest-sized graphs. No computational comparison with prior work (Bernini et al., 2024; Silvestri et al., 2025) is provided.​
- Evaluation uses only two label propagation algorithms (ANGEL, DEMON) with fixed φ=0.8. A single asymmetric test (DEMON) provides insufficient evidence for "detector-agnostic" claims. Sensitivity to merging threshold φ is unexplored.​​
- PPO selection over TRPO/A3C/SAC lacks justification. No sensitivity analysis for λ=0.1, 4 GCN layers, d_h=32, or proxy edge probability p=0.5. The threshold τ=0.5 remains constant despite Eq. 1 indicating difficulty scales with τ.
- No convergence guarantees, sample complexity bounds, or characterization of graph properties (density, modularity, clustering coefficient) affecting success rates.

**Questions:**

Questions are majorly if they could explain the weaknesses.

---

> ### Author Response · Authors · 2025-11-20
> **Response to Reviewer up1n**
>
> We thank the Reviewer for the constructive feedback. We hope to clarify their concerns below.
>
> **W1. Baselines, Comparability, and Ablations**
>
> - Comparison with Silvestri et al. (2025) & Bernini et al. (2024):
> We respectfully note that these methods are fundamentally not comparable to our setting. Both works strictly address non-overlapping community detection (partitioning). As shown in Section 4.2 (and Fig. 2 and 3), strategies that work for non-overlapping detection (like simple rewiring or naïve injection) fail catastrophically in the overlapping setting. Furthermore, neither method supports injecting external proxy nodes, focusing instead on rewiring existing connections, rendering them methodologically incompatible with our proposed framework.
>
> - Comparison with Adversarial Attacks (GANI, TDGIA):
> While GANI and TDGIA involve node injection, they are designed to attack Node Classification or Link Prediction on GNNs. Our objective is Community Hiding against structure-based detection algorithms. The loss landscapes and objectives are distinct: GANI maximizes classification error, while CMH minimizes set-similarity scores (sim) for a specific target node. Adapting GANI to optimize a black-box community similarity metric would require a complete redesign (essentially reinventing an RL approach similar to ours).
>
> - We also clarify a misunderstanding regarding the experimental setup:
> Tables 2 and 3 compare ODRL and baselines under identical proxy budgets. Specifically, we always have that $k = \beta$. Meaning that the reported gains are solely due to the learned policy we introduce in the ODRL method.
>
> **W2. Missing computational time analysis**
>
> We agree that fb-75 is a standard dataset in previous community hiding literature, but its exclusion is due to the computational cost of the environment (detection algorithm), not the agent. Overlapping community detection is computationally intensive, even given the particularly optimized *angel/demon* algorithms. For clarity, we report that a single detector call on fb-75 takes around 53 seconds on our hardware (GPU NVIDIA GeForce RTX 4090 and an AMD
> Ryzen 9 7900 12-Core CPU), with an average degree of 34, with $\beta=1.0$, we would need $\frac{53 \cdot 34 \cdot 5000}{3600 \cdot 24} \approx 104 \ \text{days}$ to train a single fb-75 community hiding model fully.
>
> To provide greater transparency regarding the computational demands of the evaluated datasets, we report the wall-clock training times for the models from Table 2 ($\beta=1.0\mu, \tau=0.5$) in the table below. This analysis will be included in the Appendix of the final paper.
>
> | Dataset    | Time (min) |
> |------------|------------|
> | `kar`      | 113.88     |
> | `words`    | 364.08     |
> | `vote`     | 504.30     |
> | `pow`      | 258.72     |
> | `nets`     | 195.00     |
>
> **W3. Limited experiments in the asymmetric setup**
>
> We acknowledge that testing on more detectors would be ideal. However, the field of overlapping community detection is fragmented, and robust, scalable implementations are scarce. We extensively tested publicly available implementations of other overlapping algorithms but found them too unstable for our experiments.
>
> **W4. Limited parameter sensitivity analysis ($k, \phi, \lambda, d_h, p, \tau$)**
>
> PPO was selected for its stability relative to A2C and its relative simplicity relative to TRPO. The network parameters were chosen via a standard coarse-grid search. We found performance to be relatively insensitive to $d_h$ (e.g., $32$ vs $64$) and prioritized a smaller model for faster training.
>
> **W5. Success Rate vs. convergence guarantees, sample complexity, or graph properties**
>
> We appreciate the reviewer raising the critical point regarding theoretical guarantees. We want to clarify that deriving global convergence bounds is constrained by the nature of the underlying detection algorithms (*angel* and *demon*), which are heuristic and combinatorial, operating as "black boxes" without closed-form gradients.
> However, to ensure training stability and theoretical groundedness despite this non-convex landscape, we explicitly chose *Proximal Policy Optimization* (PPO). PPO operates as a first-order approximation of *Trust Region Policy Optimization* (TRPO). A key theoretical advantage of TRPO is its provable guarantee of monotonic policy improvement, achieved by constraining the policy update within a specific trust region.
>
> As for the effect of our hiding method on graph properties, this evaluation framework is not well-suited for such a localized task. Although this type of analysis is common in community-hiding research, our method introduces only a minimal number of modifications (approximately one to three node injections and edge changes). For this reason, we believe it is more important to assess the impact on the overall community structure. We thank the reviewer for the suggestion and will incorporate this analysis into future work to provide a more comprehensive evaluation.

---

> ### Author Response · Authors · 2025-11-27
>
> Dear Reviewer up1n,
>
> Thank you very much again for your time and effort in reviewing our paper. To summarize our previous response and follow up respectfully, we would like to highlight the concrete steps we have taken to address your valuable comments:
>
> * **Clarified Baseline Comparability (W1)**: We explained why partition-based methods (e.g., Silvestri et al.) and GNN adversarial attacks are methodologically incompatible with the overlapping setting. We also confirmed that our reported gains are achieved under identical proxy budgets.
>
> * **Added Computational Analysis (W2)**: We provided wall-clock training times for all datasets to increase transparency, and have included these measurements in the paper's Appendix. We clarified that fb-75 was excluded due to the prohibitive cost of the detection environment (approx. 53s per call), rather than the agent's efficiency.
>
> * **Justified PPO & Theoretical Grounding (W4 & W5)**: We highlighted that PPO was chosen to ensure monotonic policy improvement in a black-box setting lacking closed-form gradients. We also noted that our method's localized edits (1-3 changes) have a negligible impact on global graph properties.
>
> We value your feedback and remain available to address any further questions you may have.

---

### Official Review · Reviewer_y2Vv · 2025-11-01

**Soundness:** 2
**Presentation:** 2
**Contribution:** 2
**Rating:** 4
**Confidence:** 3

**Summary:**

The paper introduces the problem of Community Membership Hiding (CMH) under overlapping community detection, aiming to prevent inference of sensitive affiliations in social graphs.
The authors propose a Deep Reinforcement Learning (DRL) framework, called ODRL that injects proxy nodes and optimally rewires edges to hide a target node's community membership.
The approach is evaluated on multiple real-world graph datasets and compared to heuristic and prior DRL-based baselines.
The method achieves higher F1 scores, balancing success rate and community preservation, and empirically shows generalization to unseen detection algorithms.

**Strengths:**

* **Well motivated research direction.**
The paper convincingly shows the shortcomings of naive, common strategies in the overlapping setting they consider.
The factored policy seems to be an effective approach for efficiency.


* **Promising empirical results**.
The empirical evaluation shows good results, with the proposed ODRL method generally outperforming the considered baselines with significant margin.
The approach works solidly on the asymmetric setting considered (`angel` $\to$ `demon`).

**Weaknesses:**

* **Presentation and clarity.**
The paper is very difficult to parse for someone not already very familiar with related work.
A number of methods (see next comment) are simply mentioned without any explanation or intuition of their inner workings and advantages/disadvantages.
For instance, `angel` and `demon` are presented as community detection algorithms with no details, mentioning that they are considered with a "merging threshold" of 0.8.
If not familiar with the two methods, it is difficult to attach any meaning to similar statements.
This heavily affects the clarity of the exposition and makes the contribution more difficult to judge.

* **Presentation of experimental results.**
Building up on the previous point, while I appreciate the usefulness of presenting experimental results early on, the current display of results is difficult to parse.
In Figure 2, the methods and datasets are presented without description.
For instance, "Greedy", "Louvain", and "Walktrap" are not introduced or referenced.
Similar considerations hold for Figure 3.

* **Claims of scalability.**
You discuss how your approach shows more notable advantages in large-scale graphs.
At the same time, you mention that scalability is one of the limitations your approach faces.
I think this point should be more thoroughly addressed.
The datasets you consider have a relatively small number of nodes.
And while I would not expect you to provide results on graphs with "billions of nodes", it is unclear whether the graphs you consider are the largest you could practically consider.
At the same time, you do not quantify the computational/time effort your experiments require.

* **Policy optimality.**
In your contributions, you mention that you obtain an "optimal policy" with your DRL-based approach.
First, the term "optimal" is unclearly defined for me.
Then, it is unclear to me how some notion of optimality can hold, given the large search space.

 ### Other comments

 * The acronym ODRL is never introduced.
 * In line 137, should it be $>1$ rather than $\ne 1$? Or is the case where a node belongs to no community also considered?
 * It is unclear whether the $\lambda$ in Equation 3 is the same as in line 363.

**Questions:**

* How computationally expensive is it to consider larger graphs?
* Can the proxy node injection be detected in adversarial settings? How can this be defended against?
* How does the choice of $\lambda$ affect the reward function?
* Can you expand on the claims of optimality for your policy? Is there any formal justification to the claim I may have missed?

---

> ### Author Response · Authors · 2025-11-20
> **Response to Reviewer y2Vv**
>
> We thank the Reviewer for the constructive and detailed feedback. We respond to each concern below and will incorporate all clarifications in the revised manuscript.
>
> **Q1./W2. Computational cost on larger graphs**
>
> The primary training bottleneck is the repeated invocation of the community-detection oracle. Our agent must evaluate the hiding objective many times during training, and existing overlapping detectors such as *angel* and *demon* are themselves expensive on large graphs. As noted in Section 6 and in the Limitations, our overlapping experiments failed to complete on the fb-75 snapshot due to detector runtime, not due to the intrinsic complexity of our method.
>
> Concretely, the per-episode cost $\approx$ (MDP steps per episode) $\times$ cost(community_detector). Policy forward passes are comparatively cheap: our factored policy produces only $O(|V|)$ logits per step (rather than $O(|V|^2)$), so detector calls dominate runtime.
>
> Because of the detector cost, training on massive graphs (millions of nodes) is currently impractical without further engineering: either faster detector implementations, more innovative sampling/surrogate detectors, or distributed training. We now explicitly state this and propose two concrete mitigations in the revision: (i) using learned surrogate models to approximate $f(\cdot)$ during early training, and (ii) training on graph sub-samples (inducing the agent to learn locally transferable strategies)
>
> **Q2. Detectability and defenses against proxy-node injection**
>
> Proxy injections can be detectable by a defender through some heuristics: (i) sudden appearance of multiple low-degree nodes connected only to the same target (account creation time and timestamp signals), (ii) abnormal local subgraph statistics (e.g., unusually high transitivity among new nodes or very low interaction with the rest of the graph), and (iii) feature/behavioral anomalies if node metadata is available.
>
> We reiterate the dual-use risks and recommend responsible deployment and monitoring; we already discussed this in the Ethical Considerations, and will expand it slightly to mention the detectability of proxy node injections.
>
> **Q3. Effect of $\lambda$ in the reward function?**
>
> Eq. (3) combines a binary success reward with an auxiliary shaping term that scales the relative decrease in the community-similarity measure. In our experiments, we set $\lambda=0.1$ to ensure that the success signal dominates learning while still providing practical shaping.
> To avoid ambiguity, we will rename this reward-scaling parameter to $\alpha$ in the revision.
>
> **Q4./W3. Claims of "optimality"**
>
> We agree that the “optimal policy” is misleading. We do not claim global optimality over the whole combinatorial search space. Instead, as made explicit in Eq. (7), the policy is optimal with respect to the PPO objective being optimized, i.e., it is the best policy parameters $\theta^*$ found under: $(i)$ the given MDP formulation, $(ii)$ the budget constraint $|B_u| \leq \beta$, $(iii)$ the reward in Eq. (3), and $(iv)$ the training dynamics of PPO.
>
> We will rephrase this to "policy learned to optimize our CMH objective" and make explicit that no theoretical optimality guarantees are provided, due to the size of the action space.
>
> **W1. Paper clarity**
>
> We acknowledge that several algorithms were referenced before being introduced. We will revise Sections 2 and 6 to include short, self-contained descriptions of *greedy, louvain, walktrap, angel*, and *demon*, including brief notes on why each baseline is used *(*angel* and *demon* will get one-paragraph introductions with references and a short note on the merging threshold $\phi=0.8$).
>
> **Other Comments**
>
> **C1. ODRL not defined**
>
> We thank the reviewers for bringing this editorial mistake to our attention. We will define ODRL at first occurrence and ensure consistent usage.
>
> **C2. In line 137, should it be $>1$ rather than $\neq 1$? Or is the case where a node belongs to no community also considered?**
>
> Our intended meaning is: in the non-overlapping setting, every node belongs to exactly one community ($|f(u,G)| = 1$ for all u), while in the overlapping setting, there exists at least one node u with $|f(u,G)| \neq 1$. The "$\neq 1$" form is intentional: it allows the overlapping case to include both nodes that belong to zero communities ($|f(u,G)| = 0$) and nodes that belong to multiple communities ($|f(u,G)| > 1$). Note that, for fairness, we exclude nodes that are not assigned to any community from experiments (see Section 4.3)
>
> **C3. Two distinct $\lambda$ parameters**
>
> This was an unintended notational collision: $\lambda$ appears both as a reward-scaling factor and as the GAE parameter. In the revision, we will rename the reward-scaling factor to $\alpha$ to avoid ambiguity, and keep $\lambda$ for GAE as is. Relevant lines in the manuscript that will be updated: Eq. (3) and the Training Details paragraph.

---

> > ### Comment · Reviewer_y2Vv · 2025-11-26
> >
> > Thank you for your response! As a good part of my weaknesses have to do with the presentation and clarity of your paper, I will for now keep my score and wait for the revised version of the paper.

---

> > > ### Author Response · Authors · 2025-11-27
> > >
> > > Dear Reviewer y2Vv,
> > >
> > > Thank you once again for your thoughtful feedback and for indicating your willingness to revisit your assessment after our revisions.
> > > We are writing to inform you that we have now updated the paper to address all the concerns you raised, we believe these revisions substantially improve the clarity and completeness of the manuscript. We remain available to address any further questions or concerns you may have.
> > >
> > > Thank you again for your time and constructive feedback.

---

### Author Response · Authors · 2025-12-01
**Summary of Rebuttal and Revisions for Submission 7530**

Dear Area Chair,

To help reduce your unexpected workload, we provide a concise summary of our paper's contributions and the outcome of the rebuttal. Our work, which introduces the first DRL framework for Community Membership Hiding (CMH) in overlapping communities, was recognized by all reviewers for its novelty and strong empirical performance.

We have actively engaged with the reviewers, carefully addressed every point they raised, and uploaded a revised manuscript accordingly. The improvements in clarity, contextualization, and transparency are very much the result of the reviewers' thoughtful and constructive feedback. Below is a brief overview of the strengths identified and how we incorporated their suggestions:

-    **Reviewer y2Vv** (Score: 4, Confidence: 3) praised the work as a "well-motivated research direction" with a "promising empirical evaluation" where our method outperforms baselines by a significant margin. To address their concerns regarding presentation and clarity, we extensively revised Sections 2 and 6 to provide self-contained definitions of all baselines and algorithms, and clarified that our "optimality" claims refer strictly to the learned policy within the PPO constraints rather than global combinatorial optimality.

-    **Reviewer up1n** (Score: 4, Confidence: 4) highlighted the "meaningful contribution" of our formalization and noted that ODRL’s advantage increases with graph size and complexity. We addressed their concerns regarding baselines and transparency by clarifying why partition-based methods (e.g., Silvestri et al.) are incompatible with the overlapping setting, and by adding a detailed wall-clock runtime analysis in the Appendix to distinguish agent efficiency from environment costs.

 -   **Reviewer Ji1w** (Score: 4, Confidence: 5) recognized that we address a "significant problem" using a method that "significantly outperforms baselines." We resolved their concerns regarding novelty relative to prior work by updating the Related Work section to explicitly distinguish our specific affiliation hiding objective from prior overlapping area avoidance literature, clarifying that our continuous similarity metrics prevent the false positives inherent in previous discrete approaches.

In summary, we believe that the main concerns raised, i.e., clarity, positioning with respect to disjoint-community baselines, and computational transparency, have been exhaustively addressed in the revised manuscript.

Thank you for your time and consideration.

The Authors

---

### Meta-Review · Area_Chair_46Z2 · 2025-12-18

**Summary:**

The paper studies how to prevent inference of affiliations in social community. Authors proposed an algorithm that will modify the edge. Reviewers found that the problem is interesting, but the clarity and experimental evidence need significant improvement.

**Reviewer Concerns:**

Many concerns are about experiments. While authors provided additional results, it seems that another round of full review is needed.

**Reviewer Scores:**

Reviewers may keep scores unchanged.

---

### Decision · Program_Chairs · 2026-01-26

Reject